# TRAINING WIDE RESIDUAL NETWORKS FOR DEPLOYMENT USING A SINGLE BIT FOR EACH WEIGHT

**Mark D. McDonnell** [*]
Computational Learning Systems Laboratory (`cls-lab.org`)
School of Information Technology and Mathematical Sciences
University of South Australia
Mawson Lakes, SA 5095, AUSTRALIA
`mark.mcdonnell@unisa.edu.au`

## ABSTRACT

For fast and energy-efficient deployment of trained deep neural networks on resource-constrained embedded hardware, each learned weight parameter should ideally be represented and stored using a single bit. Error-rates usually increase when this requirement is imposed. Here, we report large improvements in error rates on multiple datasets, for deep convolutional neural networks deployed with 1-bit-per-weight. Using wide residual networks as our main baseline, our approach simplifies existing methods that binarize weights by applying the sign function in training; we apply scaling factors for each layer with constant unlearned values equal to the layer-specific standard deviations used for initialization. For CIFAR-10, CIFAR-100 and ImageNet, and models with 1-bit-per-weight requiring less than 10 MB of parameter memory, we achieve error rates of 3.9%, 18.5% and 26.0% / 8.5% (Top-1 / Top-5) respectively. We also considered MNIST, SVHN and ImageNet32, achieving 1-bit-per-weight test results of 0.27%, 1.9%, and 41.3% / 19.1% respectively. For CIFAR, our error rates halve previously reported values, and are within about 1% of our error-rates for the same network with full-precision weights. For networks that overfit, we also show significant improvements in error rate by not learning batch normalization scale and offset parameters. This applies to both full precision and 1-bit-per-weight networks. Using a warm-restart learning-rate schedule, we found that training for 1-bit-per-weight is just as fast as full-precision networks, with better accuracy than standard schedules, and achieved about 98%-99% of peak performance in just 62 training epochs for CIFAR-10/100. For full training code and trained models in MATLAB, Keras and PyTorch see `https://github.com/McDonnell-Lab/1-bit-per-weight/`.

## 1 INTRODUCTION

Fast parallel computing resources, namely GPUs, have been integral to the resurgence of deep neural networks, and their ascendancy to becoming state-of-the-art methodologies for many computer vision tasks. However, GPUs are both expensive and wasteful in terms of their energy requirements. They typically compute using single-precision floating point (32 bits), which has now been recognized as providing far more precision than needed for deep neural networks. Moreover, training and deployment can require the availability of large amounts of memory, both for storage of trained models, and for operational RAM. If deep-learning methods are to become embedded in resource-constrained sensors, devices and intelligent systems, ranging from robotics to the internet-of-things to self-driving cars, reliance on high-end computing resources will need to be reduced.

To this end, there has been increasing interest in finding methods that drive down the resource burden of modern deep neural networks. Existing methods typically exhibit good performance but for the

---

[*]This work was conducted, in part, during a hosted visit at the Institute for Neural Computation, University of California, San Diego, and in part, during a sabbatical period at Consilium Technology, Adelaide, Australia.

ideal case of single-bit parameters and/or processing, still fall well-short of state-of-the-art error rates on important benchmarks.

In this paper, we report a significant reduction in the gap (see Figure 1 and Results) between Convolutional Neural Networks (CNNs) deployed using weights stored and applied using standard precision (32-bit floating point) and networks deployed using weights represented by a single-bit each.

In the process of developing our methods, we also obtained significant improvements in error-rates obtained by **full-precision** versions of the CNNs we used.

In addition to having application in custom hardware deploying deep networks, networks deployed using 1-bit-per-weight have previously been shown (Pedersoli et al., 2017) to enable significant speedups on regular GPUs, although doing so is not yet possible using standard popular libraries.

Aspects of this work was first communicated as a subset of the material in a workshop abstract and talk (McDonnell et al., 2017).

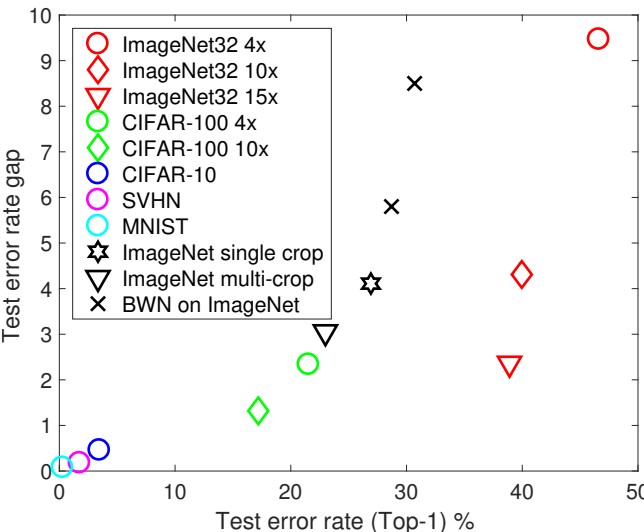

Figure 1: **Our error-rate gaps between using full-precision and 1-bit-per-weight.** All points except black crosses are data from some of our best results reported in this paper for each dataset. Black points are results on the full ImageNet dataset, in comparison with results of Rastegari et al. (2016) (black crosses). The notation 4x, 10x and 15x corresponds to network width (see Section 4).

## 1.1 RELATED WORK

### 1.1.1 RESNETS

In 2015, a new form of CNN called a "deep residual network," or "ResNet" (He et al., 2015b) was developed and used to set many new accuracy records on computer-vision benchmarks. In comparison with older CNNs such as Alexnet (Krizhevsky et al., 2012) and VGGNet (Simonyan & Zisserman, 2014), ResNets achieve higher accuracy with far fewer learned parameters and FLOPs (FLoating-point OPerations) per image processed. The key to reducing the number of parameters in ResNets was to replace "all-to-all layers" in VGG-like nets with "global-average-pooling" layers that have no learned parameters (Lin et al., 2013; Springenberg et al., 2014), while simultaneously training a much deeper network than previously. The key new idea that enabled a deeper network to be trained effectively was the introduction of so-called "skip-connections" (He et al., 2015b). Many variations of ResNets have since been proposed. ResNets offer the virtue of simplicity, and given the motivation for deployment in custom hardware, we have chosen them as our primary focus.

Despite the increased efficiency in parameter usage, similar to other CNNs the accuracy of ResNets still tends to increase with the total number of parameters; unlike other CNNs, increased accuracy can result either from deeper (He et al., 2016) or wider networks (Zagoruyko & Komodakis, 2016).

In this paper, we use Wide Residual Networks (Zagoruyko & Komodakis, 2016), as they have been demonstrated to produce better accuracy in less training time than deeper networks.

### 1.1.2 REDUCING THE MEMORY BURDEN OF TRAINED NEURAL NETWORKS

Achieving the best accuracy and speed possible when deploying ResNets or similar networks on hardware-constrained mobile devices will require minimising the total number of bits transferred between memory and processors for a given number of parameters. Motivated by such considerations, a lot of recent attention has been directed towards compressing the learned parameters (*model compression*) and reducing the precision of computations carried out by neural networks—see Hubara et al. (2016) for a more detailed literature review.

Recently published strategies for model compression include reducing the precision (number of bits used for numerical representation) of weights in deployed networks by doing the same during training (Courbariaux et al., 2015; Hubara et al., 2016; Merolla et al., 2016; Rastegari et al., 2016), reducing the number of weights in trained neural networks by pruning (Han et al., 2015; Iandola et al., 2016), quantizing or compressing weights following training (Han et al., 2015; Zhou et al., 2017), reducing the precision of computations performed in forward-propagation during inference (Courbariaux et al., 2015; Hubara et al., 2016; Merolla et al., 2016; Rastegari et al., 2016), and modifying neural network architectures (Howard et al., 2017). A theoretical analysis of various methods proved results on the convergence of a variety of weight-binarization methods (Li et al., 2017).

From this range of strategies, we are focused on an approach that simultaneously contributes two desirable attributes: (1) simplicity, in the sense that deployment of trained models immediately follows training without extra processing; (2) implementation of convolution operations can be achieved without multipliers, as demonstrated by Rastegari et al. (2016).

### 1.2 OVERALL APPROACH AND SUMMARY OF CONTRIBUTIONS

Our strategy for improving methods that enable inference with 1-bit-per-weight was threefold:

1. **State-of-the-art baseline.** We sought to begin with a baseline full-precision deep CNN variant with close to state-of-the-art error rates. At the time of commencement in 2016, the state-of-the-art on CIFAR-10 and CIFAR-100 was held by Wide Residual Networks (Zagoruyko & Komodakis, 2016), so this was our starting point. While subsequent approaches have exceeded their accuracy, ResNets offer superior simplicity, which conforms with our third strategy in this list.

2. **Make minimal changes when training for 1-bit-per-weight.** We aimed to ensure that training for 1-bit-per-weight could be achieved with minimal changes to baseline training.

3. **Simplicity is desirable in custom hardware.** With custom hardware implementations in mind, we sought to simplify the design of the baseline network (and hence the version with 1-bit weights) as much as possible without sacrificing accuracy.

### 1.2.1 CONTRIBUTIONS TO FULL-PRECISION WIDE RESNETS

Although this paper is chiefly about 1-bit-per-weight, we exceeded our objectives for the full-precision baseline network, and surpassed reported error rates for CIFAR-10 and CIFAR-100 using Wide ResNets (Zagoruyko & Komodakis, 2016). This was achieved using just 20 convolutional layers; most prior work has demonstrated best wide ResNet performance using 28 layers.

Our innovation that achieves a significant error-rate drop for CIFAR-10 and CIFAR-100 in Wide ResNets is to simply not learn the per-channel scale and offset factors in the batch-normalization layers, while retaining the remaining attributes of these layers. It is important that this is done in conjunction with exchanging the ordering of the final weight layer and the global average pooling layer (see Figure 3).

We observed this effect to be most pronounced for CIFAR-100, gaining around 3% in test-error rate. But the method is advantageous only for networks that overfit; when overfitting is not an issue, such as for ImageNet, removing learning of batch-norm parameters is only detrimental.

### 1.2.2 CONTRIBUTIONS TO DEEP CNNS WITH SINGLE-BIT WEIGHTS FOR INFERENCE

Ours is the first study we are aware of to consider how the gap in error-rate for 1-bit-per-weight compared to full-precision weights changes with full-precision accuracy across a diverse range of image classification datasets (Figure 1).

Our approach surpasses by a large margin all previously reported error rates for CIFAR-10/100 (error rates halved), for networks constrained to run with 1-bit-per-weight at inference time. One reason we have achieved lower error rates for the 1-bit case than previously is to start with a superior baseline network than in previous studies, namely Wide ResNets. However, our approach also results in smaller error rate increases relative to full-precision error rates than previously, while training requires the same number of epochs as for the case of full-precision weights.

Our main innovation is to introduce a simple fixed scaling method for each convolutional layer, that permits activations and gradients to flow through the network with minimum change in standard deviation, in accordance with the principle underlying popular initialization methods (He et al., 2015a). We combine this with the use of a warm-restart learning-rate method (Loshchilov & Hutter, 2016) that enables us to report close-to-baseline results for the 1-bit case in far fewer epochs of training than reported previously.

## 2 LEARNING A MODEL WITH CONVOLUTION WEIGHTS $\pm 1$

### 2.1 THE SIGN OF WEIGHTS PROPAGATE, BUT FULL-PRECISION WEIGHTS ARE UPDATED

We follow the approach of Courbariaux et al. (2015); Rastegari et al. (2016); Merolla et al. (2016), in that we find good results when using 1-bit-per-weight at inference time if during training we apply the sign function to real-valued weights for the purpose of forward and backward propagation, but update full-precision weights using SGD with gradients calculated using full-precision.

However, previously reported methods for training using the sign of weights either need to train for many hundreds of epochs (Courbariaux et al., 2015; Merolla et al., 2016), or use computationally-costly normalization scaling for each channel in each layer that changes for each minibatch during training, i.e. the BWN method of Rastegari et al. (2016). We obtained our results using a simple alternative approach, as we now describe.

### 2.1.1 WE SCALE THE OUTPUT OF CONV LAYERS USING A CONSTANT FOR EACH LAYER

We begin by noting that the standard deviation of the sign of the weights in a convolutional layer with kernels of size $F \times F$ will be close to 1, assuming a mean of zero. In contrast, the standard deviation of layer $i$ in full-precision networks is initialized in the method of He et al. (2015a) to $\sqrt{2/(F^2 C_{i-1})}$, where $C_{i-1}$ is the number of input channels to convolutional layer $i$, and $i = 1, .., L$, where $L$ is the number of convolutional layers and $C_0 = 3$ for RGB inputs.

When applying the sign function alone, there is a mismatch with the principled approach to controlling gradient and activation scaling through a deep network (He et al., 2015a). Although the use of batch-normalization can still enable learning, convergence is empirically slow and less effective.

To address this problem, for training using the sign of weights, we use the initialization method of He et al. (2015a) for the full-precision weights that are updated, but also introduce a layer-dependent scaling applied to the sign of the weights. This scaling has a constant unlearned value equal to the initial standard deviation of $\sqrt{2/(F^2 C_{i-1})}$ from the method of He et al. (2015a). This enables the standard deviation of forward-propagating information to be equal to the value it would have initially in full-precision networks.

In implementation, during training we multiply the sign of the weights in each layer by this value. For inference, we do this multiplication using a scaling layer following the weight layer, so that all weights in the network are stored using 0 and 1, and deployed using $\pm 1$ (see https://github.com/McDonnell-Lab/1-bit-per-weight/). Hence, custom hardware implementations would be able to perform the model's convolutions without multipliers (Rastegari et al., 2016), and significant GPU speedups are also possible (Pedersoli et al., 2017).

The fact that we scale the weights explicitly during training is important. Although for forward and backward propagation it is equivalent to scale the input or output feature maps of a convolutional layer, doing so also scales the calculated gradients with respect to weights, since these are calculated by convolving input and output feature maps. As a consequence, learning is dramatically slower unless layer-dependent learning rates are introduced to cancel out the scaling. Our approach to this is similar to the BWN method of Rastegari et al. (2016), but our constant scaling method is faster and less complex.

In summary, the only differences we make in comparison with full-precision training are as follows. Let $\mathbf{W}_i$ be the tensor for the convolutional weights in the $i$–th convolutional layer. These weights are processed in the following way only for forward propagation and backward propagation, not for weight updates:

$$\hat{\mathbf{W}}_i = \sqrt{\frac{2}{F_i^2 C_{i-1}}} \operatorname{sign}\left(\mathbf{W}_i\right), \quad i = 1, \ldots, L, \tag{1}$$

where $F_i$ is the spatial size of the convolutional kernel in layer $i$; see Figure 2.

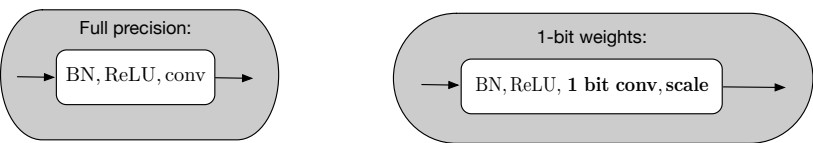

Figure 2: **Difference between our full-precision and 1-bit-per-weight networks.** The "1 bit conv" and "scale" layers are equivalent to the operations shown in Eqn. (1).

## 3 Methods common to baseline and single-bit networks

### 3.1 Network Architecture

Our ResNets use the 'post-activation' and identity mapping approach of He et al. (2016) for residual connections. For ImageNet, we use an 18-layer design, as in He et al. (2015b). For all other datasets we mainly use a 20-layer network, but also report some results for 26 layers. Each residual block includes two convolutional layers, each preceded by batch normalization (BN) and Rectified Linear Unit (ReLU) layers. Rather than train very deep ResNets, we use wide residual networks (Wide ResNets) (Zagoruyko & Komodakis, 2016). Although Zagoruyko & Komodakis (2016) and others reported that 28/26-layer networks result in better test accuracy than 22/20-layer[1] networks, we found for CIFAR-10/100 that just 20 layers typically produces best results, which is possibly due to our approach of not learning the batch-norm scale and offset parameters.

Our baseline ResNet design used in most of our experiments (see Figures 3 and 4) has several differences in comparison to those of He et al. (2016); Zagoruyko & Komodakis (2016). These details are articulated in Appendix A, and are mostly for simplicity, with little impact on accuracy. The exception is our approach of not learning batch-norm parameters.

### 3.2 Training

We trained our models following, for most aspects, the standard stochastic gradient descent methods used by Zagoruyko & Komodakis (2016) for Wide ResNets. Specifically, we use cross-entropy loss, minibatches of size 125, and momentum of 0.9 (both for learning weights, and in situations where we learn batch-norm scales and offsets). For CIFAR-10/100, SVHN and MNIST, where overfitting is evident in Wide ResNets, we use a larger weight decay of 0.0005. For ImageNet32 and full ImageNet we use a weight decay of 0.0001. Apart from one set of experiments where we added a simple extra approach called cutout, we use standard 'light' data augmentation, including randomly flipping each image horizontally with probability 0.5 for CIFAR-10/100 and ImageNet32. For the

---

[1] Two extra layers are counted when downsampling residual paths learn $1\times1$ convolutional projections.

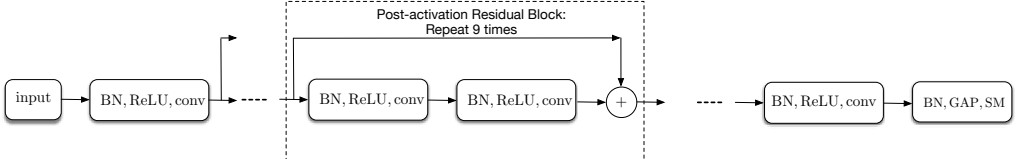

Figure 3: **Wide ResNet architecture.** The design is mostly a standard pre-activation ResNet (He et al., 2016; Zagoruyko & Komodakis, 2016). The first (stand-alone) convolutional layer ("conv") and first 2 or 3 residual blocks have $64k$ (ImageNet) or $16k$ (other datasets) output channels. The next 2 or 3 blocks have $128k$ or $32k$ output channels and so on, where $k$ is the widening parameter. The final (stand-alone) convolutional layer is a $1 \times 1$ convolutional layer that gives $N$ output channels, where $N$ is the number of classes. Importantly, this final convolutional layer is followed by batch-normalization ("BN") prior to global-average-pooling ("GAP") and softmax ("SM"). The blocks where the number of channels double are downsampling blocks (details are depicted in Figure 4) that reduce each spatial dimension in the feature map by a factor of two. The rectified-linear-unit ("RELU") layer closest to the input is optional, but when included, it is best to learn the BN scale and offset in the subsequent layer.

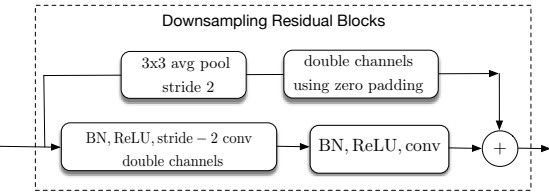

Figure 4: **Downsampling blocks in Wide ResNet architecture.** As in a standard pre-activation ResNet (He et al., 2016; Zagoruyko & Komodakis, 2016), downsampling (stride-2 convolution) is used in the convolutional layers where the number of output channels increases. The corresponding downsampling for skip connections is done in the same residual block. Unlike standard pre-activation ResNets we use an average pooling layer ("avg pool" ) in the residual path when downsampling.

same 3 datasets plus SVHN, we pad by 4 pixels on all sides (using random values between 0 and 255) and crop a $32 \times 32$ patch from a random location in the resulting $40 \times 40$ image. For full ImageNet, we scale, crop and flip, as in He et al. (2015b). We did not use any image-preprocessing, i.e. we did not subtract the mean, as the initial BN layer in our network performs this role, and we did not use whitening or augment using color or brightness. We use the initialization method of He et al. (2015a).

We now describe important differences in our training approach compared to those usually reported.

### 3.2.1 BATCH-NORM SCALES & OFFSETS ARE NOT LEARNED FOR CIFAR-10/100, SVHN

When training on CIFAR-10/100 and SVHN, in all batch-norm layers (except the first one at the input when a ReLU is used there), we do not learn the scale and offset factor, instead initializating these to 1 and 0 in all channels, and keeping those values through training. Note that we also do not learn any biases for convolutional layers.

The usual approach to setting the moments for use in batch-norm layers in inference mode is to keep a running average through training. When not learning batch-normalization parameters, we found a small benefit in calculating the batch-normalization moments used in inference only after training had finished. We simply form as many minibatches as possible from the full training-set, each with the same data augmentation as used during training applied, and pass these through the trained network, averaging the returned moments for each batch.

For best results when using this method using matconvnet, we found it necessary to ensure the $\epsilon$ parameter that is used to avoid divisions by zero is set to $1 \times 10^{-5}$; this $\epsilon$ is different to the way it is used in keras and other libraries.

### 3.3 OUR NETWORK'S FINAL WEIGHT LAYER IS A $1 \times 1$ CONVOLUTIONAL LAYER

A significant difference to the ResNets of He et al. (2016); Zagoruyko & Komodakis (2016) is that we exchange the ordering of the global average pooling layer and the final weight layer, so that our final weight layer becomes a $1 \times 1$ convolutional layer with as many channels as there are classes in the training set. This design is not new, but it does seem to be new to ResNets: it corresponds to the architecture of Lin et al. (2013), which originated the global average pooling concept, and also to that used by Springenberg et al. (2014).

As with all other convolutional layers, we follow this final layer with a batch-normalization layer; the benefits of this in conjunction with not learning the batch-normalization scale and offset are described in the Discussion section.

#### 3.3.1 WE USE A WARM-RESTART LEARNING-RATE SCHEDULE

We use a 'warm restarts' learning rate schedule that has reported state-of-the-art Wide ResNet results (Loshchilov & Hutter, 2016) whilst also speeding up convergence. The method constantly reduces the learning rate from 0.1 to $1 \times 10^{-4}$ according to a cosine decay, across a certain number of epochs, and then repeats across twice as many epochs. We restricted our attention to a maximum of 254 epochs (often just 62 epochs, and no more than 30 for ImageNet32) using this method, which is the total number of epochs after reducing the learning rate from maximum to minimum through 2 epochs, then 4, 8, 16, 32, 64 and 128 epochs. For CIFAR-10/100, we typically found that we could achieve test error rates after 32 epochs within 1-2% of the error rates achievable after 126 or 254 epochs.

#### 3.3.2 EXPERIMENTS WITH CUTOUT FOR CIFAR-10/100

In the literature, most experiments with CIFAR-10 and CIFAR-100 use simple "standard" data augmentation, consisting of randomly flipping each training image left-right with probability 0.5, and padding each image on all sides by 4 pixels, and then cropping a $32 \times 32$ version of the image from a random location. We use this augmentation, although with the minor modification that we pad with uniform random integers between 0 and 255, rather than zero-padding.

Additionally, we experimented with "cutout" (Devries & Taylor, 2017). This involves randomly selecting a patch of each raw training image to remove. The method was shown to combine with other state-of-the-art methods to set the latest state-of-the-art results on CIFAR-10/100 (see Table 1). We found better results using larger cutout patches for CIFAR-100 than those reported by Devries & Taylor (2017); hence for both CIFAR-10 and CIFAR-100 we choose patches of size $18 \times 18$. Following the method of Devries & Taylor (2017), we ensured that all pixels are chosen for being included in a patch equally frequently throughout training by ensuring that if the chosen patch location is near the image border, the patch impacts on the image only for the part of the patch inside the image. Differently to Devries & Taylor (2017), as for our padding, we use uniform random integers to replace the image pixel values in the location of the patches. We did not apply cutout to other datasets.

## 4 RESULTS

We conducted experiments on six databases: four databases of $32 \times 32$ RGB images—CIFAR-10, CIFAR-100, SVHN and ImageNet32—and the full ILSVRC ImageNet database (Russakovsky et al., 2015), as well as MNIST (LeCun et al., 1998). Details of the first three $32 \times 32$ datasets can be found in many papers, e.g. (Zagoruyko & Komodakis, 2016). ImageNet32 is a downsampled version of ImageNet, where the training and validation images are cropped using their annotated bounding boxes, and then downsampled to $32 \times 32$ (Chrabaszcz et al., 2017); see http://image-net.org/download-images. All experiments were carried out on a single GPU using MATLAB with GPU acceleration from MatConvNet and cuDNN.

We report results for Wide ResNets, which (except when applied to ImageNet) are $4\times$ and $10\times$ wider than baseline ResNets, to use the terminology of Zagoruyko & Komodakis (2016), where the baseline has 16 channels in the layers at the first spatial scale. We use notation of the form 20-10 to denote Wide ResNets with 20 convolutional layers and 160 channels in the first spatial scale, and hence width $10\times$. For the full ImageNet dataset, we use 18-layer Wide ResNets with 160 channels in the first spatial scale, but given the standard ResNet baseline is width 64, this corresponds to width $2.5\times$ on this dataset (Zagoruyko & Komodakis, 2016). We denote these networks as 18-2.5.

Table 1 lists our top-1 error rates for CIFAR-10/100; C10 indicates CIFAR-10, C100 indicates CIFAR-100; the superscript $^+$ indicates standard crop and flip augmentation; and $^{++}$ indicates the use of cutout. Table 2 lists error rates for SVHN, ImageNet32 (I32) and full ImageNet; we did not use cutout on these datasets. Both top-1 and top-5 results are tabulated for I32 and ImageNet. For ImageNet, we provide results for single-center-crop testing, and also for multi-crop testing. In the latter, the decision for each test image is obtained by averaging the softmax output after passing through the network 25 times, corresponding to crops obtained by rescaling to 5 scales as described by He et al. (2015b), and from 5 random positions at each scale. Our full-precision ImageNet error rates are slightly lower than expected for a wide ResNet according to the results of Zagoruyko & Komodakis (2016), probably due to the fact we did not use color augmentation.

Table 1: Test-set error-rates for our approach applied to CIFAR-10 and CIFAR-100.

| Weights | ResNet | Epochs | Params | C10$^+$ | C100$^+$ | C10$^{++}$ | C100$^{++}$ |
|---------|--------|--------|--------|---------|----------|------------|-------------|
| 32-bits | 20-4 | 126 | 4.3M | 5.02 | 21.53 | 4.39 | 20.48 |
| 32-bits | 20-10 | 254 | 26.8M | 4.22 | 18.76 | 3.46 | 17.19 |
| 32-bits | 26-10 | 254 | 35.6M | 4.23 | 18.63 | 3.54 | 17.22 |
| 32-bits | 20-20 | 126 | 107.0M | - | 18.14 | - | - |
| 1-bit | 20-4 | 126 | 4.3M | 6.13 | 23.87 | 5.34 | 23.74 |
| 1-bit | 20-10 | 254 | 26.8M | 4.72 | 19.35 | 3.92 | 18.51 |
| 1-bit | 26-10 | 254 | 35.6M | 4.46 | 18.94 | 3.41 | 18.50 |
| 1-bit | 20-20 | 126 | 107.0M | - | 18.81 | - | - |

Table 2: Test-set error-rates for our approach applied to SVHN, ImageNet32, and ImageNet.

| Weights | ResNet | Epochs | Params | SVHN | I32 | ImageNet |
|---------|--------|--------|--------|------|-----|----------|
| 32-bits | 20-4 | 30 | 4.5M | 1.75 | 46.61 / 22.91 | - |
| 32-bits | 20-10 | 30 | 27.4M | - | 39.96 / 17.89 | - |
| 32-bits | 20-15 | 30 | 61.1M | - | 38.90 / 17.03 | - |
| 32-bits | 18-2.5 | 62 | 70.0M | - | - | Single crop: 26.92 / 9.20
Multi-crop: 22.99 / 6.91 |
| 1-bit | 20-4 | 30 | 4.5M | 1.93 | 56.08 / 30.88 | - |
| 1-bit | 20-10 | 30 | 27.4M | - | 44.27 / 21.09 | - |
| 1-bit | 26-10 | 62 | $\sim$ 36M | - | 41.36 / 18.93 | - |
| 1-bit | 20-15 | 30 | 61.1M | - | 41.26 / 19.08 | - |
| 1-bit | 18-2.5 | 62 | 70.0M | - | - | Single crop: 31.03 / 11.51
Multi-crop: 26.04 / 8.48 |

Table 3 shows comparison results from the original work on Wide ResNets, and subsequent papers that have reduced error rates on the CIFAR-10 and CIFAR-100 datasets. We also show the only results, to our knowledge, for ImageNet32. The current state-of-the-art for SVHN without augmentation is $1.59\%$ (Huang et al., 2016), and with cutout augmentation is $1.30\%$ (Devries & Taylor, 2017). Our full-precision result for SVHN ($1.75\%$) is only a little short of these even though we used only a $4\times$ ResNet, with less than 5 million parameters, and only 30 training epochs.

Table 4 shows comparison results for previous work that trains models by using the sign of weights during training. Additional results appear in Hubara et al. (2016), where activations are quantized, and so the error rates are much larger.

Inspection of Tables 1 to 4 reveals that our baseline full-precision $10\times$ networks, when trained with cutout, surpass the performance of deeper Wide ResNets trained with dropout. Even without the use

Table 3: Test error rates for networks with less than 40M parameters, sorted by CIFAR-100.

| Method | # params | C10 | C100 | I32 Top-1 / Top-5 |
|---|---|---|---|---|
| WRN 22-10 (Zagoruyko & Komodakis, 2016) | 27M | 4.44 | 20.75 | - |
| **1-bit weights WRN 20-10 (This Paper)** | **27M** | **4.72** | **19.35** | **44.27 / 21.09** |
| WRN 28-10 (Chrabaszcz et al., 2017) | 37M | - | - | 40.97 / 18.87 |
| **Full precision WRN 20-10 (This Paper)** | **27M** | **4.22** | **18.76** | **39.96 / 17.89** |
| **1-bit weights WRN 20-10 + cutout (This Paper)** | **27M** | **3.92** | **18.51** | - |
| WRN 28-10 + cutout (Devries & Taylor, 2017) | 34M | 3.08 | 18.41 | - |
| WRN 28-10 + dropout (Zagoruyko & Komodakis, 2016) | 37M | 3.80 | 18.30 | - |
| ResNeXt-29, 8×64d Xie et al. (2016) | 36M | 3.65 | 17.77 | - |
| **Full precision WRN 20-10 + cutout (This Paper)** | **27M** | **3.46** | **17.19** | - |
| DenseNets (Huang et al., 2016) | 26M | 3.46 | 17.18 | - |
| Shake-shake regularization (Gastaldi, 2017) | 26M | 2.86 | 15.97 | - |
| Shake-shake + cutout (Devries & Taylor, 2017) | 26M | 2.56 | 15.20 | - |

Table 4: Test error rates using 1-bit-per-weight at test time and propagation during training.

| Method | C10 | C100 | SVHN | ImageNet |
|---|---|---|---|---|
| BC (Courbariaux et al., 2015) | 8.27 | - | 2.30 | - |
| Weight binarization (Merolla et al., 2016) | 8.25 | - | - | - |
| BWN - Googlenet (Rastegari et al., 2016) | 9.88 | - | - | 34.5 / 13.9 (full ImageNet) |
| VGG+HWGQ (Cai et al., 2017) | 7.49 | - | - | - |
| BC with ResNet + ADAM (Li et al., 2017) | 7.17 | 35.34 | - | 52.11 (full ImageNet) |
| BW with VGG (Cai et al., 2017) | - | - | - | 34.5 (full ImageNet) |
| **This Paper: single center crop** | **3.41** | **18.50** | **1.93** | **41.26 / 19.08 (ImageNet32)** 
 **31.03 / 11.51 (full ImageNet)** |
| **This Paper: 5 scales, 5 random crops** | - | - | - | **26.04 / 8.48 (full ImageNet)** |

of cutout, our 20-10 network surpasses by over 2% the CIFAR-100 error rate reported for essentially the same network by Zagoruyko & Komodakis (2016) and is also better than previous Wide ResNet results on CIFAR-10 and ImageNet32. As elaborated on in Section 5.3, this improved accuracy is due to our approach of not learning the batch-norm scale and offset parameters.

For our 1-bit networks, we observe that there is always an accuracy gap compared to full precision networks. This is discussed in Section 5.1.

Using cutout for CIFAR-10/100 reduces error rates as expected. In comparison with training very wide $20\times$ ResNets on CIFAR-100, as shown in Table 1, it is more effective to use cutout augmentation in the $10\times$ network to reduce the error rate, while using only a quarter of the weights.

Figure 5 illustrates convergence and overfitting trends for CIFAR-10/100 for 20-4 Wide ResNets, and a comparison of the use of cutout in 20-10 Wide ResNets. Clearly, even for width-4 ResNets, the gap in error rate between full precision weights and 1-bit-per-weight is small. Also noticeable is that the warm-restart method enables convergence to very good solutions after just 30 epochs; training longer to 126 epochs reduces test error rates further by between 2% and 5%. It can also be observed that the 20-4 network is powerful enough to model the CIFAR-10/100 training sets to well over 99% accuracy, but the modelling power is reduced in the 1-bit version, particularly for CIFAR-100. The reduced modelling capacity for single-bit weights is consistent with the gap in test-error rate performance between the 32-bit and 1-bit cases. When using cutout, training for longer gives improved error rates, but when not using cutout, 126 epochs suffices for peak performance.

Finally, for MNIST and a $4\times$ wide ResNet without any data augmentation, our full-precision method achieved 0.71% after just 1 epoch of training, and 0.28% after 6 epochs, whereas our 1-bit method achieved 0.81%, 0.36% and 0.27% after 1, 6 and 14 epochs. In comparison, 1.29% was reported for the 1-bit-per-weight case by Courbariaux et al. (2015), and 0.96% by Hubara et al. (2016).

## 4.1 ABLATION STUDIES

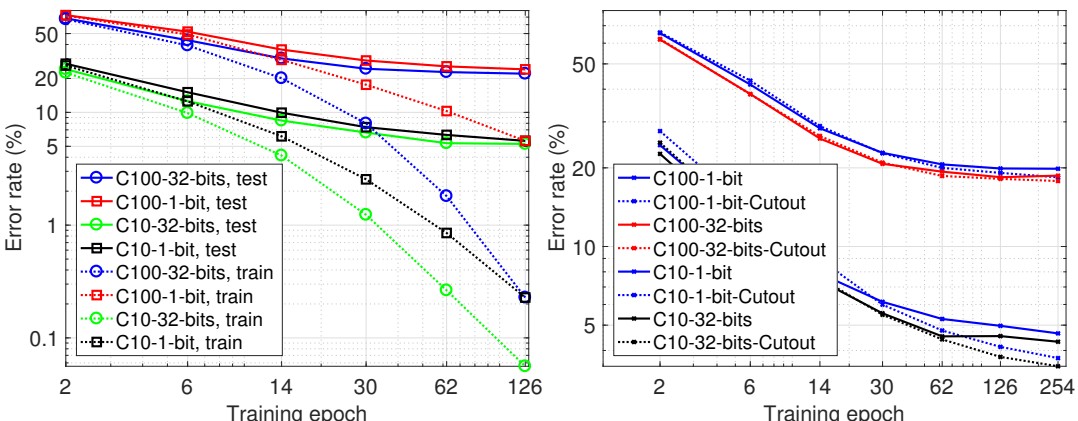

Figure 5: **Convergence through training.** Left: Each marker shows the error rates on the test set and the training set at the end of each cycle of the warm-restart training method, for 20-4 ResNets (less than 5 million parameters). Right: each marker shows the test error rate for 20-10 ResNets, with and without cutout. C10 indicates CIFAR-10, and C100 indicates CIFAR-100.

For CIFAR-10/100, both our full-precision Wide ResNets and our 1-bit-per-weight versions benefit from our method of not learning batch-norm scale and offset parameters. Accuracy in the 1-bit-per-weight case also benefits from our use of a warm-restart training schedule (Loshchilov & Hutter, 2016). To demonstrate the influence of these two aspects, in Figure 6 we show, for CIFAR-100, how the test error rate changes through training when either or both of these methods are not used. We did not use cutout for the purpose of this figure. The comparison learning-rate-schedule drops the learning rate from 0.1 to 0.01 to 0.001 after 85 and 170 epochs. It is clear that our methods lower the final error rate by around 3% absolute by not learning the batch-norm parameters. The warm-restart method enables faster convergence for the full-precision case, but is not significant in reducing the error rate. However, for the 1-bit-per-weight case, it is clear that for best results it is best to both use warm-restart, and to not learn batch-norm parameters.

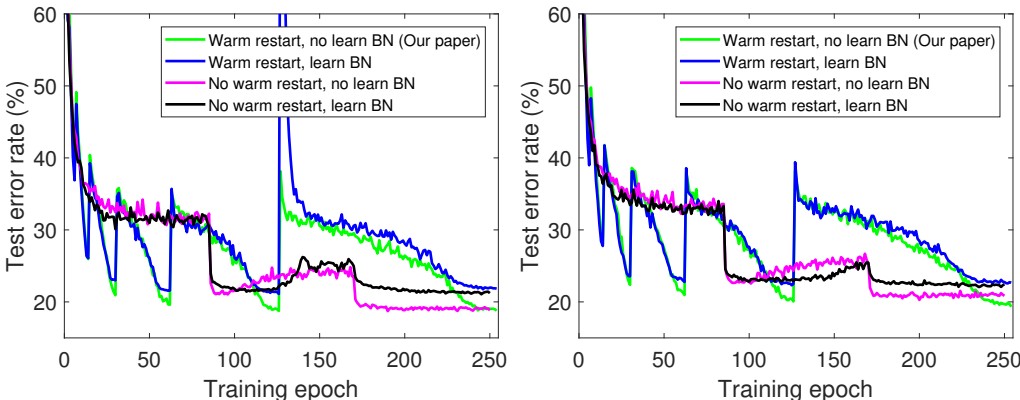

Figure 6: **Influence of warm-restart and not learning BN gain and offset.** Left: full-precision case. Right: 1-bit-per-weight case.

## 5 DISCUSSION

### 5.1 THE ACCURACY GAP FOR 1-BIT VS 32-BIT WEIGHTS

It is to be expected that a smaller error rate gap will result between the same network using full-precision and 1-bit-per-weight when the test error rate on the full precision network gets smaller.

Indeed, Tables 1 and 2 quantify how the gap in error rate between the full-precision and 1-bit-per-weight cases tends to grow as the error-rate in the full-precision network grows.

To further illustrate this trend for our approach, we have plotted in Figure 1 the gap in Top-1 error rates vs the Top-1 error rate for the full precision case, for some of the best performing networks for the six datasets we used. Strong conclusions from this data can only be made relative to alternative methods, but ours is the first study to our knowledge to consider more than two datasets. Nevertheless, we have also plotted in Figure 1 the error rate and the gap reported by Rastegari et al. (2016) for two different networks using their BWN 1-bit weight method. The reasons for the larger gaps for those points is unclear, but what is clear is that better full-precision accuracy results in a smaller gap in all cases.

A challenge for further work is to derive theoretical bounds that predict the gap. How the magnitude of the gap changes with full precision error rate is dependent on many factors, including the method used to generate models with 1-bit-per-weight. For high-error rate cases, the loss function throughout training is much higher for the 1-bit case than the 32-bit case, and hence, the 1-bit-per-weight network is not able to fit the training set as well as the 32-bit one. Whether this is because of the loss of precision in weights, or due to the mismatch in gradients inherent is propagating with 1-bit weights and updating full-precision weights during training is an open question. If it is the latter case, then it is possible that principled refinements to the weight update method we used will further reduce the gap. However, it is also interesting that for our 26-layer networks applied to CIFAR-10/100, that the gap is much smaller, despite no benefits in the full precision case from extra depth, and this also warrants further investigation.

## 5.2    COMPARISON WITH THE BWN METHOD

Our approach differs from the BWN method of Rastegari et al. (2016) for two reasons. First, we do not need to calculate mean absolute weight values of the underlying full precision weights for each output channel in each layer following each minibatch, and this enables faster training. Second, we do not need to adjust for a gradient term corresponding to the appearance of each weight in the mean absolute value. We found overall that the two methods work equally effectively, but ours has two advantages: faster training, and fewer overall parameters. As a note we found that the method of Rastegari et al. (2016) also works equally effectively on a per-layer basis, rather than per-channel.

We also note that the focus of Rastegari et al. (2016) was much more on the case that combines single-bit activations and 1-bit-per-weight than solely 1-bit-per-weight. It remains to be tested how our scaling method compares in that case. It is also interesting to understand whether the use of batch-norm renders scaling of the sign of weights robust to different scalings, and whether networks that do not use batch-norm might be more sensitive to the precise method used.

## 5.3    THE INFLUENCE OF NOT LEARNING BATCH-NORMALIZATION PARAMETERS

The unusual design choice of not learning the batch normalization parameters was made for CIFAR-10/100, SVHN and MNIST because for Wide ResNets, overfitting is very evident on these datasets (see Figure 5); by the end of training, typically the loss function becomes very close to zero, corresponding to severe overfitting. Inspired by label-smoothing regularization (Szegedy et al., 2015) that aims to reduce overconfidence following the softmax layer, we hypothesized that imposing more control over the standard deviation of inputs to the softmax might have a similar regularizing effect. This is why we removed the final all-to-all layer of our ResNets and replaced it with a $1 \times 1$ convolutional layer followed by a batch-normalization layer prior to the global average pooling layer. In turn, not learning a scale and offset for this batch-normalization layer ensures that batches flowing into the softmax layer have standard deviations that do not grow throughout training, which tends to increase the entropy of predictions following the softmax, which is equivalent to lower confidence (Guo et al., 2017).

After observing success with these methods in $10 \times$ Wide ResNets, we then observed that learning the batch-norm parameters in other layers also led to increased overfitting, and increased test error rates, and so removed that learning in all layers (except the first one applied to the input, when ReLU is used there).

As shown in Figure 6, there are significant benefits from this approach, for both full-precision and 1-bit-per-weight networks. It is why, in Table 3, our results surpass those of Zagoruyko & Komodakis (2016) on effectively the same Wide ResNet (our 20-10 network is essentially the same as the 22-10 comparison network, where the extra 2 layers appear due to the use of learned $1 \times 1$ convolutional projections in downsampling residual paths, whereas we use average pooling instead).

As expected from the motivation, we found our method is not appropriate for datasets such as ImageNet32 where overfitting is not as evident, in which case learning the batch normalization parameters significantly reduces test error rates.

## 5.4 COMPARISON WITH SQUEEZENET

Here we compare our approach with SqueezeNet (Iandola et al., 2016), which reported significant memory savings for a trained model relative to an AlexNet. The SqueezeNet approach uses two strategies to achieve this: (1) replacing many 3x3 kernels with 1x1 kernels; (2) deep compression (Han et al., 2015).

Regarding SqueezeNet strategy (1), we note that SqueezeNet is an all-convolutional network that closely resembles the ResNets used here. We experimented briefly with our 1-bit-per-weight approach in many all-convolutional variants—e.g. plain all-convolutional (Springenberg et al., 2014), SqueezeNet (Iandola et al., 2016), MobileNet (Howard et al., 2017), ResNeXT (Xie et al., 2016)—and found its effectiveness relative to full-precision baselines to be comparable for all variants. We also observed in many experiments that the total number of learned parameters correlates very well with classification accuracy. When we applied a SqueezeNet variant to CIFAR-100, we found that to obtain the same accuracy as our ResNets for about the same depth, we had to increase the width until the SqueezeNet had approximately the same number of learned parameters as the ResNet. We conclude that our method therefore reduces the model size of the baseline SqueezeNet architecture (i.e. when no deep compression is used) by a factor of 32, albeit with an accuracy gap.

Regarding SqueezeNet strategy (2), the SqueezeNet paper reports deep compression (Han et al., 2015) was able to reduce the model size by approximately a factor of 10 with no accuracy loss. Our method reduces the same model size by a factor of 32, but with a small accuracy loss that gets larger as the full-precision accuracy gets smaller. It would be interesting to explore whether deep compression might be applied to our 1-bit-per-weight models, but our own focus is on methods that minimally alter training, and we leave investigation of more complex methods for future work.

Regarding SqueezeNet performance, the best accuracy reported in the SqueezeNet paper for ImageNet is 39.6% top-1 error, requiring 4.8MB for the model's weights. Our single-bit-weight models achieve better than 33% top-1 error, and require 8.3 MB for the model's weights.

## 5.5 LIMITATIONS AND FURTHER WORK

In this paper we focus only on reducing the precision of weights to a single-bit, with benefits for model compression, and enabling of inference with very few multiplications. It is also interesting and desirable to reduce the computational load of inference using a trained model, by carrying out layer computations in very few numbers of bits (Hubara et al., 2016; Rastegari et al., 2016; Cai et al., 2017). Facilitating this requires modifying non-linear activations in a network from ReLUs to quantized ReLUs, or in the extreme case, binary step functions. Here we use only full-precision calculations. It can be expected that combining our methods with reduced precision processing will inevitably increase error rates. We have addressed this extension in a forthcoming submission.

### ACKNOWLEDGMENTS

This work was supported by a Discovery Project funded by the Australian Research Council (project number DP170104600). Discussions and visit hosting by Gert Cauwenberghs and Hesham Mostafa, of UCSD, and André van Schaik and Runchun Wang of Western Sydney University are gratefully acknowledged, as are discussions with Dr Victor Stamatescu and Dr Muhammad Abul Hasan of University of South Australia.

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

## A    DIFFERENCES FROM STANDARD RESNETS IN OUR BASELINE

### A.1    OUR NETWORK DOWNSAMPLES THE RESIDUAL PATH USING AVERAGE POOLING

For skip connections between feature maps of different sizes, we use zero-padding for increasing the number of channels as per option 1 of He et al. (2015b). However, for the residual pathway, we use average pooling using a kernel of size $3 \times 3$ with stride 2 for downsampling, instead of typical lossy downsampling that discards all pixel values in between samples.

### A.2    WE USE BATCH NORMALIZATION APPLIED TO THE INPUT LAYER

The literature has reported various options for the optimal ordering, usage and placement of BN and ReLU layers in residual networks. Following He et al. (2016); Zagoruyko & Komodakis (2016), we precede convolutional layers with the combination of BN followed by ReLU.

However, different to He et al. (2016); Zagoruyko & Komodakis (2016), we also insert BN (and optional ReLU) immediately after the input layer and before the first convolutional layer. When the

optional ReLU is used, unlike all other batch-normalization layers, we enable learning of the scale and offset factors. This first BN layer enables us to avoid doing any pre-processing on the inputs to the network, since the BN layer provides necessary normalization. When the optional ReLU is included, we found that the learned offset ensures the input to the first ReLU is never negative.

In accordance with our strategy of simplicity, all weight layers can be thought of as a block of three operations in the same sequence, as indicated in Figure 3. Conceptually, batch-norm followed by ReLU can be thought of as a single layer consisting of a ReLU that adaptively changes its centre point and positive slope for each channel and relative to each mini-batch.

We also precede the global average pooling layer by a BN layer, but do not use a ReLU at this point, since nonlinear activation is provided by the softmax layer. We found including the ReLU leads to differences early in training but not by the completion of training.

### A.3 OUR FIRST CONV LAYER HAS AS MANY CHANNELS AS THE FIRST RESIDUAL BLOCK

The Wide ResNet of Zagoruyko & Komodakis (2016) is specified as having a first convolutional layer that always has a constant number of output channels, even when the number of output channels for other layers increases. We found there is no need to impose this constraint, and instead always allow the first layer to share the same number of output channels as all blocks at the first spatial scale. The increase in the total number of parameters from doing this is small relative to the total number of parameters, since the number of input channels to the first layer is just 3. The benefit of this change is increased simplicity in the network definition, by ensuring one fewer change in the dimensionality of the residual pathway.

## B COMPARISON OF RESIDUAL NETWORKS AND PLAIN CNNS

We were interested in understanding whether the good results we achieved for single-bit weights were a consequence of the skip connections in residual networks. We therefore applied our method to plain all-convolutional networks identical to our $4\times$ residual networks, except with the skip connections removed. Initially, training indicated a much slower convergence, but we found that altering the initial weights standard deviations to be proportional to 2 instead of $\sqrt{2}$ helped, so this was the only other change made. The change was also applied in Equation (1).

As summarized in Figure 7, we found that convergence remained slower than our ResNets, but there was only a small accuracy penalty in comparison with ResNets after 126 epochs of training. This is consistent with the findings of He et al. (2015b) where only ResNets deeper than about 20 layers showed a significant advantage over plain all-convolutional networks. This experiment, and others we have done, support our view that our method is not particular to ResNets.

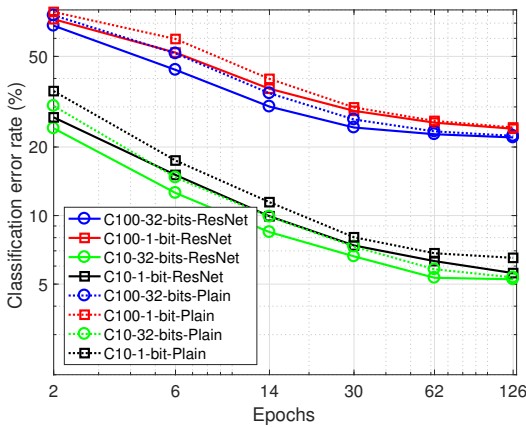

Figure 7: **Residual networks compared with all-convolutional networks.** The data in this figure is for networks with width $4\times$, i.e. with about 4.3 million learned parameters.

