# OpenReview forum: "Training wide residual networks for deployment using a single bit for each weight"
_ICLR.cc/2018/Conference — Accept (Poster)_

### Official Review · AnonReviewer1 · 2017-11-27
**Mixed ideas**

**Rating:** 6
**Confidence:** 4

**Review:**

This paper introduces several ideas: scaling, warm-restarting learning rate, cutout augmentation.

I would like to see detailed ablation studies: how the performance is influenced by the warm-restarting learning rates, how the performance is influenced by cutout. Is the scaling scheme helpful for existing single-bit algorithms?

Question for Table 3: 1-bit WRN 20-10 (this paper) outperforms WRN 22-10 with the same #parameters on C100. I would like to see more explanations.

---

> ### Author Response · Authors · 2017-12-31
> **Response to AnonReviewer1**
>
> Thankyou for your comments and questions.
>
> ***
>
> Reviewer Comment: “I would like to see detailed ablation studies: how the performance is influenced by the warm-restarting learning rates, how the performance is influenced by cutout”
>
> Author Response:
>
> We already separated out the influence of cutout in the original submission. We only used cutout for CIFAR 10/100, and Table 1 showed separate columns for results without cutout (indicated by superscript +) and those with cutout (indicated by superscript ++).  Figure 5 (right panel) shows how the use of cutout influences convergence.
>
> We did not originally provide comparisons with and without warm-restart, because the benefits of the warm-restart method (both faster convergence and better accuracy) for CIFAR 10/100 has already been established by Loshchilov and Hutter (2016) who compared the approach with a more typical schedule with a learning rate of 0.1/0.01/0.001 for 80/80/80 epochs. However, the reviewer has a point that the comparison has not previously been done for the case of single-bit-weights and hence we have now conducted some experiments.
>
> Author actions:
>
> 1.	We have added a section in Results called “Ablation Studies” and included a new figure for CIFAR-100. The figure highlights that the warm-restart method does not provide a significant accuracy benefit for the full-precision case but does in the single-bit-weights case. The figure also shows a comparison of learning and not learning the batch-norm offsets and gains, in response to another question by this Reviewer, responded to below.
>
> ***
>
> Reviewer question: “Is the scaling scheme helpful for existing single-bit algorithms?
>
> Author Response: Our Section 2.1 describes how our approach builds on and enables the improvement of existing single-bit algorithms. Our new Section 4.1 shows how our use of warm-restart accelerates convergence, and provides  best accuracy, especially for CIFAR-10.  Our Section 5.2 discusses the specific case of how our method compares with Rastegari et al (2016).
>
> Author Action: we have added Section 4.1 and revised Section 5.2.
>
> ***
>
> Reviewer question: “Question for Table 3: 1-bit WRN 20-10 (this paper) outperforms WRN 22-10 with the same #parameters on C100. I would like to see more explanations.”
>
> Author response: In this initial submission, we only explained this in general terms in Section 5.3 as being a result of our approach described in 3.2.1. So we agree that the reviewer has a point. We did not highlight this aspect very much in the original submission, nor explain it in the specific cases tabulated, as we wanted the central emphasis to be on our single-bit-weight results. However, on reflection, improvements to the baseline approach are surely of interest to the community and worth emphasis.
>
> For the specific case mentioned by the Reviewer, we remark that our 20-10 network is essentially the same as the 22-10 comparison network, where the extra 2 conv layers appear due to the use of learnt 1x1 convolutional projections in downsampling residual paths, whereas we use average pooling instead.
>
> To directly answer the question, there is one single factor that enabled us to significantly lower the error rate for the width-10 wide ResNet architecture for CIFAR, which is that we turn off the learning of the batch norm parameters, as we found this reduces overfitting.
>
> Author actions:
>
> 1.	We have now highlighted this contribution in the abstract.
> 2.	We have now highlighted the specific case mentioned in the Discussion in Section 5.3.
> 3.	We have added results in a new Section (“Ablation studies”) that show how the test error rate changes through training with and without learning of the batch-norm scale and offset.

---

### Official Review · AnonReviewer2 · 2017-11-27
**Solid work**

**Rating:** 6
**Confidence:** 3

**Review:**

The authors propose to train neural networks with 1bit weights by storing and updating full precision weights in training, but using the reduced 1bit version of the network to compute predictions and gradients in training. They add a few tricks to keep the optimization numerically efficient. Since right now more and more neural networks are deployed to end users, the authors make an interesting contribution to a very relevant question.

The approach is precisely described although the text sometimes could be a bit clearer (for example, the text contains many important references to later sections).

The authors include a few other methods for comparision, but I think it would be very helpful to include also some methods that use a completely different approach to reduce the memory footprint. For example, weight pruning methods sometimes can give compression rates of around 100 while the 1bit methods by definition are limited to a compression rate of 32. Additionally, for practical applications, methods like weight pruning might be more promising since they reduce both the memory load and the computational load.

Side mark: the manuscript has quite a few typos.

---

> ### Author Response · Authors · 2018-01-01
> **Response to AnonReviewer2**
>
> Thankyou for your comments.
>
> ***
>
> Reviewer Comment:  “could be a bit clearer… the text contains many important references to later sections”
>
> Author Action: We have edited the text to improve this aspect.
>
> ***
>
> Reviewer Comment: “I think it would be very helpful to include also some methods that use a completely different approach to reduce the memory footprint…. for practical applications, methods like weight pruning might be more promising since they reduce both the memory load and the computational load”
>
> Author Response:
>
> As well as reducing model size, our approach is strongly motivated by significantly reducing computational load by a different approach to reducing parameter number. The key point is that performing convolutions using 1-bit weights can be implemented using adders rather than multipliers. Removing the need for multipliers offers enormous benefits in terms of chip size, speed and power consumption in custom digital hardware implementations of trained networks, and also offers substantial speedups even if implemented on GPUs. This has been demonstrated by Rastegari et al in “ XNOR-Net: ImageNet Classification Using Binary Convolutional Neural Networks” (arxiv: 1603.05279, 2016).
>
> Existing pruning methods do not automatically offer the opportunity to avoid use of multiplications, since the un-pruned parameters are learned using full precision. It remains an open question beyond the scope of the current submission to determine whether pruning can be successfully applied to 1-bit models like ours to in turn reduce the number of parameters.
>
> Question to Reviewer:  we have been unable to find methods that reduce the size of all-convolutional networks by a factor of 100. This magnitude of reduction is, to our knowledge, only available in networks with very large fully-connected layers, such as AlexNet and VGGnet. For example, one submission to ICLR 2018 “To Prune, or Not to Prune: Exploring the Efficacy of Pruning for Model Compression” uses pruning applied to an Inception network that reduces the number of non-zero parameters from 27M to 3M, which is a factor of 9x. Can you please clarify if you know of papers that achieve this scale of pruning in all-convolutional  networks such as ResNets?
>
> Author Action: we have added comments strengthening our motivation of reducing the use of multiplications, and added to our discussion of pruning in the prior work and discussion.
>
> ***
>
> Reviewer Comment: “the manuscript has quite a few typos..”
>
> Author Action: we have carefully reviewed the entire manuscript and corrected typos.

---

> > ### Comment · AnonReviewer2 · 2018-01-12
> > **Response to response**
> >
> > Thankyou for your response and the updated manuscript, especially for detailing your motivation for using 1bit convolutions. Just out of curiosity: Do you happen to have rough numbers how large the speedup on regular GPUs is when implementing 1bit convolutions as you suggested instead of using standard GPU convolutions with 1bit numbers?
> >
> > Regarding the compression rate of 100 that I cited: I was in fact referring to VGGNet and I primarily tried to make the point that it would be useful to compare your method to completely different approaches. With respect to this, I appreciate the newly added section on SqueezeNet.

---

> > > ### Author Response · Authors · 2018-01-23
> > > **Response to response to response**
> > >
> > > Regarding speedup on GPUs: we did all our work using the standard approach of using 32-bit GPU implementations to simulate the 1-bit case, in which case there's no speedup.  The reason is that custom GPU code needs to be written, presumably in cuda; we didn't need to do this to conduct our study.
> > >
> > > However, we found a paper submitted to this ICLR: "Espresso: Efficient Forward Propagation for Binary Deep Neural Networks" (https://openreview.net/forum?id=Sk6fD5yCb) that reports a 5x speed increase for optimized GPU code for binary networks applied to CIFAR.

---

> > > > ### Comment · AnonReviewer2 · 2018-01-23
> > > > **Thanks**
> > > >
> > > > Ah, that is an impressive speedup! Thanks! (I would suggest citing this in the paper since it serves nicely at making the motivation of using 1bit weights obvious. But I leave that decision up to you).

---

> > > > > ### Author Response · Authors · 2018-01-23
> > > > > **Re: thanks**
> > > > >
> > > > > Good suggestion - we will do that at the next opportunity for revision.

---

### Official Review · AnonReviewer3 · 2017-11-28
**a single bit for each weight**

**Rating:** 6
**Confidence:** 4

**Review:**

The paper trains wide ResNets for 1-bit per weight deployment.
The experiments are conducted on CIFAR-10, CIFAR-100, SVHN and ImageNet32.

+the paper reads well
+the reported performance is compelling

Perhaps the authors should make it clear in the abstract by replacing:
"Here, we report methodological innovations that result in large reductions in error rates across multiple datasets for deep convolutional neural networks deployed using a single bit for each weight"
with
"Here, we report methodological innovations that result in large reductions in error rates across multiple datasets for wide ResNets deployed using a single bit for each weight"

I am curious how the proposed approach compares with SqueezeNet (Iandola et al.,2016) in performance and memory savings.

---

> ### Author Response · Authors · 2017-12-31
> **response to AnonReviewer3**
>
> Thankyou for your comments.
>
> ***
>
> Reviewer comment: "+the reported performance is compelling":
>
> Author Response: To reinforce this aspect, since initial submission we have found the following ways to surpass the performance we initially reported:
>
> 1. We now have conducted experiments on the full Imagenet dataset and have surpassed all previously published results for a single-bit per weight. Indeed, we provide the first report, to our knowledge, of a top-5 error rate under 10% for this case.
> 2. For Imagenet32, we realised that the weight decay we used was set to the larger CIFAR value of 0.0005. We repeated our experiments with the usual Imagenet value of 0.0001 and achieved improved results.
> 3. For experiments on CIFAR with cutout, we realised our previous experiments did not uniformly sample all pixels for cutout; after fixing we achieved further reduced error rates.
> 4. We have also completed experiments with CIFAR 10/100 for ResNets with depth 26. We found the extra layers provided no benefit for the full-precision case, but a small advantage in the single-bit case.
>
> Author Actions: We have updated the results tables in the revised manuscript, modified our descriptions of the use of CutOut, clarified our weight-decay values, and added comments in the Discussion section comparing aspects of the enhanced results.
>
> ***
>
> Reviewer Comment: “Perhaps the authors should make it clear in the abstract…”
>
> Author Response: You have a point that our experiments in the main text were all on wide ResNets. This followed from our strategy to commence with a near state-of-the-art baseline. However, our training approach is general and not specific to ResNets. For example, we provided some results for all-conv-nets in the Appendix B on the final page.
>
> Author Actions: To improve clarity as suggested, we have added the phrase "Using depth-20 wide residual networks as our main baseline" to our revised manuscript, but have retained the term "deep convolutional neural networks."
>
> ***
>
> Reviewer Comment: “I am curious how the proposed approach compares with SqueezeNet (Iandola et al.,2016) in performance and memory savings. “
>
> Author Response: The Squeezenet paper focuses on memory savings relative to AlexNet. It uses two strategies to produce a memory-saving smaller model than an AlexNet: (1) replacing many 3x3 kernels with 1x1 kernels; (2) deep compression.
>
> Regarding Squeezenet memory-saving strategy (1), we note that SqueezeNet is an all-convolutional network. We tried our single-bit-weights approach in many all-convolutional variants (e.g. plain all-conv, Squeezenet, MobileNet, ResNeXt) and found its effectiveness relative to full-precision baselines to be comparable for all variants. We also observed in many experiments that the total number of learnt parameters correlates very well with classification accuracy. When we applied a SqueezeNet variant to CIFAR-100, we found that to obtain the same accuracy as our ResNets, we had to increase the "width" until the SqueezeNet had approximately the same number of learnt parameters as the ResNet. We conclude that our method therefore reduces the model size of the baseline SqueezeNet architecture (i.e. when no deep compression is used) by a factor of 32, albeit with an accuracy gap.
>
> Regarding SqueezeNet memory-saving strategy (2), the SqueezeNet paper reports that Deep Compression reduces the model size by approximately a factor of 10 with no accuracy loss. Our method reduces the same model size by a factor of 32, but with a small accuracy loss that typically becomes larger as the full-precision accuracy gets smaller. It would certainly be interesting to explore whether Deep-Compression might be applied to our 1-bit models, but our own focus is on methods that minimally alter training, and we leave investigation of more complex methods for future work.
>
> Regarding SqueezeNet performance, the best accuracy reported in the SqueezenNet paper is 39.6% top-1 error, requiring 4.8MB for the model’s weights. Our single-bit-weight models achieve better than 33% top-1 error, and require 8.3 MB for the model’s weights.
>
> Author Actions: We added these comments to a new subsection in the Discussion section of our paper.

---

### Author Response · Authors · 2018-01-04
**Summary of changes to paper following review**

1. Note first that changes made in response to specific comments from reviewers are written in our response to each reviewer. In summary, significant changes in this regard include:
1a. the addition of a figure for an ablation study, as requested by a reviewer
1b. the addition of a section comparing our work to SqueezeNet in the Discussion, as requested by a reviewer
1c. more emphasis on our improvements to full-precision training, by not learning batch-norm parameters

2.  We also updated all results, finished experiments on ImageNet, updating the abstract accordingly.

---

### Decision · Program_Chairs · 2018-01-29
**ICLR 2018 Conference Acceptance Decision**

**Decision:**

Accept (Poster)

**Comment:**

The paper presents a way of training 1bit wide resnet to reduce the model footprint while maintaining good performance. The revisions added more comparisons and discussions, which make it much better. Overall, the committee feels this work will bring value to the conference.